# Synthesis, Biological Evaluation, and In Silico Studies of Novel Aminated Xanthones as Potential p53-Activating Agents

**DOI:** 10.3390/molecules24101975

**Published:** 2019-05-22

**Authors:** Agostinho Lemos, Ana Sara Gomes, Joana B. Loureiro, Pedro Brandão, Andreia Palmeira, Madalena M. M. Pinto, Lucília Saraiva, Maria Emília Sousa

**Affiliations:** 1Laboratory of Organic and Pharmaceutical Chemistry, Department of Chemical Sciences, Faculty of Pharmacy, University of Porto, Rua de Jorge Viterbo Ferreira, 228, 4050-313 Porto, Portugal; up201002662@ff.up.pt (A.L.); pedrocgbrandao@gmail.com (P.B.); andreiapalmeira@gmail.com (A.P.); madalena@ff.up.pt (M.M.M.P.); 2UCIBIO/REQUIMTE, Laboratory of Microbiology, Department of Biological Sciences, Faculty of Pharmacy, University of Porto, Rua de Jorge Viterbo Ferreira, 228, 4050-313 Porto, Portugal; anasarag4@gmail.com (A.S.G.); up201407524@ff.up.pt (J.B.L.); 3CIIMAR-Interdisciplinary Centre of Marine and Environmental Research, University of Porto, Novo Edificio do Terminal de Cruzeiros do Porto de Leixões, Avenida General Norton de Matos, S/N, 4450-208 Matosinhos, Portugal

**Keywords:** antitumor activity, computational docking, MDM2-p53 interaction, xanthones, yeast-based assays

## Abstract

Xanthone scaffold has been regarded as an attractive chemical tool in the search for bioactive molecules with antitumor activity, and in particular two xanthone derivatives, 12-hydroxy-2,2-dimethyl-3,4-dihydro-2*H*,6*H*-pyrano [3,2-*b*]xanthen-6-one (**4**) and 3,4-dimethoxy-9-oxo-9*H*-xanthene-1-carbaldehyde (**5**), were described as a murine double minute 2 (MDM2)-p53 inhibitor and a TAp73 activator, respectively. The xanthone **5** was used as a starting point for the construction of a library of 3,4-dioxygenated xanthones bearing chemical moieties of described MDM2-p53 inhibitors. Eleven aminated xanthones were successfully synthesized and initially screened for their ability to disrupt the MDM2-p53 interaction using a yeast cell-based assay. With this approach, xanthone **37** was identified as a putative p53-activating agent through inhibition of interaction with MDM2. Xanthone **37** inhibited the growth of human colon adenocarcinoma HCT116 cell lines in a p53-dependent manner. The growth inhibitory effect of xanthone **37** was associated with the induction of G1-phase cell cycle arrest and increased protein expression levels of p53 transcriptional targets. These results demonstrated the potential usefulness of coupling amine-containing structural motifs of known MDM2-p53 disruptors into a 3,4-dioxygenated xanthone scaffold in the design of novel and potent p53 activators with antitumor activity and favorable drug-like properties. Moreover, in silico docking studies were performed in order to predict the binding poses and residues involved in the potential MDM2-p53 interaction.

## 1. Introduction

The tumor suppressor p53 acts as a transcription factor, triggering the expression of multiple downstream target genes that play a pivotal role in the regulation of cell cycle arrest, apoptosis, senescence, and DNA repair [1,2,3,4,5,6]. Upon cellular stress stimuli, the activation of the p53 pathway may compromise the tumorigenesis, circumventing the proliferation of damaged cells with oncogenic potential. Inactivation of the p53 function represents one of the most common events in human cancers [7]. In approximately half of all human cancer cells, the p53 inactivation is caused by mutations within the p53 gene (*TP53*) or by post-translational modifications of its gene product [8]. In those tumors retaining the wild type (wt) p53 function, the tumor suppressor activity of p53 can be abolished by multiple mechanisms [9], mainly by direct interaction between p53 and its primary endogenous inhibitor, the murine double minute 2 (MDM2) protein [10,11]. The oncoprotein MDM2 physically interacts with the amino-terminal domain of p53 and negatively regulates its activity by blocking p53-mediated transactivation [12,13], by inducing the export of p53 from nucleus to the cytoplasm [14], and by promoting E3 ubiquitin ligase-mediated p53 degradation via the proteasomal pathway [15,16]. Targeting the MDM2-p53 interaction with small molecules has been regarded an attractive approach for the reactivation of the p53 pathway in wt p53-expressing tumors [17,18]. Since the seminal work of Vassilev et al. reporting the discovery of nutlin-3A (**1**, Figure 1) as the first inhibitor of MDM2-p53 interaction [19], an impressive number of MDM2-p53 disruptors from a huge diversity of chemical families have been identified in recent years, entering some of them into clinical trials [20,21].

Xanthones represent a privileged class of compounds with a multiplicity of biological activities, in particular with antitumor activity against several human tumor cell lines [22,23,24,25,26,27,28,29,30,31,32,33,34,35,36]. Intriguingly, our research group has demonstrated that the antitumor activity of some derivatives can be correlated to the disruption of the MDM2-p53 complex and the activation of a p53-dependent pathway [30,31]. In fact, implementing a yeast-based assay, the naturally occurring xanthones gambogic acid (**2**, Figure 1) and α-mangostin (**3**, Figure 1) were found to revert the inhibitory effect of MDM2 on p53-induced growth inhibition and on p53 transcriptional activity [30]. The synthetic xanthone 12-hydroxy-2,2-dimethyl-3,4-dihydro-2*H*,6*H*-pyrano [3,2-*b*]xanthen-6-one (**4**, Figure 1) was identified as a potential MDM2-p53 disruptor, using an in silico virtual screening strategy against MDM2 protein and a yeast-based assay [31]. Its molecular mechanism was corroborated in MDM2-overexpressing human tumor cell lines with wt p53. In fact, xanthone **4** selectively upregulated the p53 transcriptional activity, increasing the protein expression levels of p53-target genes in HCT116 human colon adenocarcinoma cells [31]. Nevertheless, xanthone **4** is highly hydrophobic and poorly water soluble, which could limit its in vivo use. Interestingly, Chen and co-workers explored the antitumor activity of xanthone **4** either in combination or hybridized with 5,6-dimethylxanthone-4-acetic acid (DMXAA) [37], a xanthone derivative with tumor-vascular disrupting activity. A very recent study reported the discovery of 3,4-dimethoxy-9-oxo-9*H*-xanthene-1-carbaldehyde (**5**, Figure 1) as a new TAp73 activator via disruption of its interaction with both MDM2 and mutant p53 [36].

Herein, based on the 3,4-oxygenated pattern of substitution of xanthones **4** and **5** (Figure 2A), and on chemical moieties derived from known MDM2-p53 disruptors, namely the molecules **1** [19], **6** [38], **7** [39], and **8** [40] (Figure 2B), we followed a molecular hybridization approach for the construction of novel 3,4-oxygenated xanthone derivatives (Figure 2D). In the present work, we opted to synthesize a library of aminated xanthones (Figure 2D) via a strategy of reductive amination of the xanthone **5** with a set of commercially available amine precursors (**9**–**19**) containing structural motifs of MDM2-p53 disruptors (Figure 2C). The introduction of amine moieties in the xanthone scaffold may contribute to a potential enhancement of drug-like properties, a critical parameter for the identification of promising drug candidates. The ability of the synthesized ligands to disrupt the MDM2-p53 interaction and to activate a p53-dependent pathway was further investigated using the previously established yeast-based assay and in human cancer cell lines. In silico docking studies were performed in order to predict the binding poses and residues involved in the potential MDM2-p53 interaction.

## 2. Results and Discussion

### 2.1. Chemistry

#### 2.1.1. Synthesis of 3,4-dimethoxy-9-oxo-9*H*-xanthene-1-carbaldehyde (**5**)

In the present work, xanthone **5** was used as a starting precursor for the synthesis of a set of aminated xanthones with a 3,4-dioxygenated pattern of substitution. Recently, the multi-step synthesis of compound **5** has been reported [36] as consisting in: (i) a Friedel-Crafts acylation of 1,2,3-trimethoxybenzene (**20**) with the commercially available reagent 2-methoxybenzoyl chloride (**21**); (ii) an intramolecular nucleophilic aromatic substitution of (2-hydroxy-3,4-dimethoxyphenyl)(2-methoxyphenyl)methanone (**22**) and consecutive regioselective demethylation of the methoxy group at position C-4 under conventional heating; (iii) a Duff formylation at position C-1 of 4-hydroxy-3-methoxy-9*H*-xanthen-9-one (**23**); and (iv) a methylation of the hydroxyl group at position C-4 of 4-hydroxy-3-methoxy-9-oxo-9*H*-xanthene-1-carbaldehyde (**24**) (Scheme 1A). The difficulty in achieving a regioselective demethylation at position C-4 necessary for Duff formylation at position C-1 demanded the planning of an optimized synthetic route more appropriate for scaling-up. We hypothesized that the implementation of an analog precursor of **20** with a suitable substituent at position C-1 might provide a useful alternative for the synthesis of **5**, avoiding the troublesome demethylation and Duff formylation processes. Herein, the commercially available 1,2,3-trimethoxy-5-methylbenzene (**25**) was selected for the Friedel-Crafts acylation with the acyl chloride **21** and the Lewis acid aluminum chloride to furnish the intermediate (2-hydroxy-3,4-dimethoxy-6-methylphenyl)(2-methoxyphenyl)methanone (**26**). Subsequent intramolecular nucleophilic aromatic substitution followed by elimination of methanol under basic conditions and microwave (MW) irradiation provided the 3,4-dimethoxy-1-methyl-9*H*-xanthen-9-one (**27**) in good yield quantity. This two-step methodology for the construction of xanthone analogues of **27** has been previously described in the literature [32,41,42,43]. With the application of MW methodology in the basic cyclization of **26** instead of the conventional heating employed in the synthesis of **23**, the reaction time was reduced from 48 h to 6 h. Furthermore, xanthone **27** was obtained in higher reaction yield under MW irradiation (63% yield (**27**) vs. 38% yield (**23**)). Subsequently, the Vohl-Ziegler bromination of **27**, using dibenzoyl peroxide (DBP) and *N*-bromosuccinimide (NBS) as a radical initiator and a brominating agent, respectively, afforded the 1-(dibromomethyl)-3,4-dimethoxy-9*H*-xanthen-9-one (**28**) as the major product. A solvolytic displacement of bromine atoms of the xanthone **28** was effectively attained using 1-butyl-3-methylimidazolium tetrafluoroborate and water, under conventional heating, and furnished xanthone **5** in 64% yield quantity (Scheme 1B). Of note, the two-step procedure of Vohl-Ziegler bromination and subsequent solvolytic displacement required lower reaction times (4.5 h (2 steps from **27** to **5**) vs. 54 h (2 steps from **23** to **5**) and allowed the preparation of xanthone **5** in higher reaction yields (51% yield (2 steps from **27** to **5**) vs. 28% yield (2 steps from **23** to **5**)), compared to the Duff formylation/methylation pathway. Overall, the replacement of starting precursor **20** by the methylated derivative **25** at position C-1 of the aromatic ring was critical for the successful optimization of the preparation of xanthone **5**.

#### 2.1.2. Synthesis of a Library of Aminated Xanthones (**30**–**40**)

A library of aminated xanthones **30**–**40** with a 3,4-dioxygenated pattern of substitution was obtained through a one-pot reductive amination of xanthone **5** with the appropriate amine precursors **9**–**19** (Scheme 2). In one-pot reductive aminations, the selection of the reducing agent is critical for their efficiency, since a suitable agent must reduce the Schiff base intermediate selectively over the carbonylated compounds. As an attempt to investigate the best reaction conditions, a solid-supported reactant containing borohydride (MP-BH_4_) combined with methanol were initially employed, since it was successfully used in the reductive amination of a xanthone derivative (data not shown). When these conditions were applied to the reductive amination of **5**, the starting precursor was not completely consumed and the alcohol derivative 1-(hydroxymethyl)-3,4-dimethoxy-9*H*-xanthen-9-one (**29**) was the major product formed. The replacement of the above-mentioned conditions by the milder reducing agent sodium triacetoxyborohydride (STAB) and the solvent tetrahydrofuran allowed a total consumption of the xanthone **5**, favoring the one-pot amination/reduction pathway. Under the reaction conditions described in the Scheme 2, eleven aminated xanthones **30**–**40** were successfully synthesized in moderate to good yields (11 examples, 39–73% yields).

The structure elucidation of the xanthones **5** and **27**–**40** was established on the basis of high-resolution mass spectrometry (HRMS) (Appendix A) and nuclear magnetic resonance (NMR) techniques (Appendix A). The ^13^C-NMR assignments were determined by bidimensional heteronuclear single quantum coherence (HSQC) and heteronuclear multiple bond correlation (HMBC) experiments (Appendix A).

### 2.2. Biological Activity Evaluation

#### 2.2.1. Effect of Synthesized Xanthone Derivatives **30**–**40** on the MDM2-p53 Interaction Using A Yeast-Based Screening Assay

Our group has previously developed a yeast-based screening assay to search for potential inhibitors of the MDM2-p53 interaction [30,31,44,45,46]. In this yeast cell system, the expression of human wt p53 induces a growth inhibition proportional to the degree of its activity (Figure 3, p53, dimethyl sulfoxide (DMSO)). On the other hand, this p53 inhibitory effect is reversed by yeast co-expression of human MDM2 (Figure 3, MDM2-p53, DMSO). This established yeast screening assay was used to study the potential inhibitory effect of 1-20 µM of xanthone derivatives **30**–**40** on the MDM2-p53 interaction. Figure 3 represents the maximal effect achieved with xanthone derivatives **30**–**40** at 10 µM. The results showed that only the xanthone **37** caused a significant reversion of the inhibitory effect of MDM2 on p53 (77.01 ± 7.35% of reversion, *n* = 3), with an almost complete reestablishment of the p53 growth inhibitory effect (63.55 ± 4.26% vs. 69.22 ± 2.23%; Figure 3, DMSO-treated p53-expressing yeast vs. xanthone **37**-treated MDM2-p53-expressing yeast). It should be noted that xanthone **37** did not interfere with the growth of control yeast (transformed with empty vectors) or yeast expressing human p53 alone (data not shown). As such, these results indicated that the xanthone **37** behaved as a potential p53-activating agent through inhibition of interaction with MDM2, and it was, therefore, selected for further investigation in human cancer cells.

#### 2.2.2. Evaluation of the Antitumor Activity of Xanthone 37 in Human Cancer Cell Lines

Using the sulforhodamine B (SRB) assay, it was confirmed that xanthone **37** inhibited the growth of human HCT116 p53^+/+^ colon cancer cells, with an IC_50_ (concentration that causes 50% growth inhibition) value of 8.67 ± 0.59 µM (*n* = 4), and of the MDM2-overexpressing human HepG2 liver carcinoma cells, with an IC_50_ value of 18.95 ± 0.39 µM (*n* = 4), after 48 h treatment.

To evaluate the dependency of the antitumor activity of xanthone **37** on the p53 pathway, we next determined the impact of xanthone **37** on the colony forming ability of HCT116 p53^+/+^ cells and on the respective p53-knockout (HCT116 p53^−/−^), using a colony formation assay (Figure 4A). The results showed a significant reduction of the growth inhibitory activity of xanthone **37**, at 3–5 μM in HCT116 p53^+/+^ cells, demonstrating a p53-dependent antitumor effect of xanthone **37**. Accordingly, 10 and 20 µM of xanthone **37** caused a G1-phase cell cycle arrest in HCT116 p53^+/+^ cells, but not in HCT116 p53^−/−^ cells (Figure 4B) after 24 h treatment. It is of note that apoptosis analysis was also investigated by Annexin-V assay; nevertheless, apoptotic events were not detected at 10 and 20 µM of xanthone **37**, after 24 and 48 h treatment. In accordance with these results, we also observed that xanthone **37** upregulated the protein expression levels of MDM2, p53, and p21, in HCT116 p53^+/+^ cells for 24 h treatment (Figure 4C).

Altogether, these results indicated that xanthone **37** had an in vitro p53-dependent antitumor activity mediated by induction of cell cycle arrest.

### 2.3. In Slico Studies

It is common knowledge that the amino-terminal (residues **18**–**26**) p53 α-helical peptide interacts with a deep hydrophobic cleft within the amino-terminal domain of MDM2 [47]. The MDM2 binding site consists of a large and a small pocket that interact with Phe19/Trp23 and Leu26 of p53, respectively (Figure 5A) [48]. The crystal structure of the MDM2 complexed with the transactivation domain of p53 (Protein Data Bank (PDB) id 1YCR [49]) allowed the visual inspection of the interactions between the Phe19, Trp23, and Leu26 residues of p53 and their respective pockets. Hydrogen-bonding interactions are established between the indole group of Trp23 and carbonyl backbone of Leu54, and between the NH backbone of Phe19 and the carbamoyl group of Gln72. Additional interactions are established between p53 and Leu57, Gly58, Ile61, Met62, Val75, Val93, and His96 residues of MDM2 (Figure 5B).

The 1.6 Å X-ray crystal structure of the small-molecule inhibitor of MDM2-p53 interaction **1** in complex with MDM2 (PDB id 4HG7 [50]) was selected for further molecular docking studies of the aminated xanthone derivatives **30**–**40** onto MDM2 protein, using the MDM2-p53 disruptors **1**–**4** as positive controls. In this work, AutoDock Vina [51] was the software chosen to predict docking conformations, as it has been described as being the best software in predicting crystallographic MDM2-p53 inhibitor poses (root-mean-square deviation (RMSD) < 1.0 Å) by redocking tests [52]. Docking scores for the most stable binding poses of xanthones **30**–**40** on MDM2 were determined using the AutoDock Vina software [51]. The predicted docking scores for known MDM2-p53 inhibitors (positive controls **1**–**4**) ranged from −7.7 to −8.0 kcal·mol^−1^ (Table 1). The most stable binding conformation of the synthesized xanthones **30**–**40** exhibited docking scores from −5.9 to −7.4 kcal· mol^−1^, therefore being in the same range of binding affinity as the positive controls (Table 1).

Amongst the tested compounds, the xanthone **37** was identified as the most active compound in the yeast-based screening assay. In addition, the predicted free energy values for the most stable binding pose of xanthone **37** were more negative than the majority of the remaining ligands. Therefore, the xanthone **37** was further analyzed in terms of docking poses and residues potentially involved in the interaction with MDM2. Figure 6A shows the most stable conformation of xanthone **37** as suggested by the docking protocol. The xanthone **37** was predicted to interact within the p53-binding site, mimicking the key hydrophobic residues of p53 when bound to MDM2 (docking score of −7.3 kcal·mol^−1^). Similarly to the binding mode of **1** within the p53-binding site (Figure 6B), no hydrogen-bonding interactions between the xanthone **37** and the MDM2 residues are predicted to be established. The *p*-fluorobenzylamino group of xanthone **37** occupies the Trp23 pocket and makes π-stacking interactions with Phe91 residue, amide-π interactions with the Leu57 and Gly58 residues, and CH-π interactions with the Leu54, Leu57, Gly58, Ile61, Val93, and Ile99 residues. Interestingly, the docking conformation of *p*-fluorobenzylamino group and the *p*-chlorophenyl ring of **1** are identical and superimposable (Figure 6A,B). The methoxy groups of the xanthone scaffold are oriented to the Leu26 pocket and make CH-CH interactions with the Leu54 residue. The adopted scaffold orientation of the xanthone **37** in the hydrophobic cleft where p53 is supposed to bind allows additional interactions to be formed, strengthening the binding of xanthone **37** to MDM2. For example, apart from the Gly58 residue of MDM2, the non-substituted aromatic ring of the xanthone scaffold is predicted to establish *π*-stacking interactions with the Phe55 residue, and amide-*π* and CH-*π* interactions with the Gln59 residue. The establishment of additional interactions within the MDM2 protein that are not observed in the MDM2-p53 binding have been described for a considerable number of high-affinity p53-activating agents [17]. The visual inspection of the xanthone **37** also suggests that an enhancement of its binding affinity towards MDM2 protein can be achieved by molecular modifications in the xanthone scaffold of **37** in order to optimize its interaction with the amino acid residues surrounding the Phe19 pocket.

## 3. Materials and Methods

### 3.1. Chemistry

All reagents and solvents were purchased from Sigma Aldrich (Sigma-Aldrich Co. Ltd., Gillinghan, UK) and no further purification process was implemented. Solvents were evaporated using a rotary evaporator under reduced pressure, Buchi Waterchath B-480. Microwave (MW) reactions were performed using an Ethos MicroSYNTH 1600 Microwave Labstation from Milestone (Thermo Unicam, Portugal). The internal reaction temperature was controlled by a fiber optic probe sensor. All reactions were monitored by thin-layer chromatography (TLC) carried out on precoated plates with 0.2 mm of thickness using Merck silica gel 60 (GF_254_) with appropriate mobile phases. Compounds **30**–**40** were easily detectable at 254 nm or 365 nm and after revelation with a solution of ninhydrin in ethanol 3 mg·mL^−1^ (activated by heat).

Flash column chromatography using silica gel 60 (0.040–0.063 mm, Merck, Darmstadt, Germany), flash cartridge chromatography (GraceResolv^®^, Grace Company, Deerfield, IL, USA), and Discovery^®^ DSC-SCX SPE cationic exchange cartridge (Grace Company, Deerfield, IL, USA) were used in the purification of the synthesized compounds. Melting points (m.p.) were measured in a Köfler microscope (Wagner and Munz, Munich, Germany) and are uncorrected. ^1^H- and ^13^C-nuclear magnetic resonance (NMR) spectra were recorded at the University of Aveiro, Department of Chemistry in CDCl_3_ or DMSO-*d*_6_ (Deutero GmbH, Ely, UK) at room temperature on a Bruker Avance 300 spectrometer (300.13 MHz for ^1^H and 75.47 MHz for ^13^C, Bruker Biosciences Corporation, Billerica, MA, USA). Chemical shifts are expressed in δ (ppm) values relative to tetramethylsilane (TMS) as an internal reference. Coupling constants are reported in hertz (Hz). ^13^C-NMR assignments were made by bidimensional heteronuclear single quantum coherence (HSQC) and heteronuclear multiple bond correlation (HMBC) NMR experiments (long-range C, H coupling constants were optimized to 7 Hz) or by comparison with the assignments of similar molecules. High-resolution mass spectroscopy (HRMS) spectra were measured on a Bruker FTMS APEX III mass spectrometer (Bruker Corporation, Billerica, MA, USA) and recorded as electrospray ionization (ESI) mode in Centro de Apoio Cientifico e Tecnolóxico á Investigación (CACTI, University of Vigo, Pontevedra, Spain). The following compounds were synthesized and purified by the described procedures.

#### 3.1.1. Synthesis of 3,4-dimethoxy-1-methyl-9*H*-xanthen-9-one (**27**)

Xanthone **27** (9.34 g, 63% yield) was synthesized from 1,2,3-trimethoxy-5-methylbenzene (**25**) and characterized according to the previously described procedure [42].

#### 3.1.2. Synthesis of 1-(dibromomethyl)-3,4-dimethoxy-9*H*-xanthen-9-one (**28**)

A mixture of **27** (2.52 g, 9.32 mmol), *N*-bromosuccinimide (3.32 g, 18.7 mmol), and dibenzoyl peroxide (0.68 g, 2.8 mmol) in carbon tetrachloride (25 mL) was refluxed (85 °C) for 2 h. The reaction was monitored using *n*-hexane/ethyl acetate in a proportion of 8:2. Once the reaction was completed, the resulting orange suspension was cooled at 0 °C and stirred for 30 min in an ice bath. The solid was filtered and washed with cold carbon tetrachloride. The filtrate was evaporated under reduced pressure and further purified by flash column chromatography (SiO_2_, *n*-hexane/ethyl acetate in gradient) to obtain the **28** as a white powder.

For 1-(dibromomethyl)-3,4-dimethoxy-9*H*-xanthen-9-one (**28**): White powder (3.14 g, 80% yield); melting point (m.p.) 125–127 °C. ^1^H-NMR (CDCl_3_, 300.13 MHz): *δ* = 8.91 (1H, s, H-1), 8.30 (1H, dd, *J* = 8.0 and 1.6 Hz, H-8), 7.77 (1H, s, H-2), 7.73 (1H, ddd, *J* = 8.4, 7.0, and 1.6 Hz, H-6), 7.54 (1H, d, *J* = 8.4 Hz, H-5), 7.39 (1H, ddd, *J* = 7.5, H-7), 4.11 (3H, s, 4-OCH_3_), 4.05 (3H, s, 3-OCH_3_) ppm; ^13^C-NMR (CDCl_3_, 75.47 MHz): *δ* = 177.8 (C-9), 156.5 (C-3), 155.0 (C-10a), 150.4 (C-4a), 139.5 (C-4), 137.6 (C-1), 135.0 (C-6), 126.9 (C-8), 124.3 (C-7), 122.0 (C-8a), 117.6 (C-5), 111.9 (C-2), 110.8 (C-9a), 61.9 (3-OCH_3_), 56.6 (4-OCH_3_), 39.2 (C-1′) ppm. HRMS (ESI^+^): *m/z* [C_16_H_12_Br_2_O_5_ + H]^+^ calcd. for [C_16_H_13_Br_2_O_5_]: 426.91751; found: 426.91709.

#### 3.1.3. Synthesis of 3,4-dimethoxy-9-oxo-9*H*-xanthene-1-carbaldehyde (**5**)

Xanthone **28** (3.14 g, 7.34 mmol) was added to a solution of 1-butyl-3-methylimidazolium tetrafluoroborate and water (6 mL, 5:1). The mixture was heated at 100 °C with stirring for 2.5 h. The reaction was monitored using *n*-hexane/ethyl acetate in a proportion of 7:3. After completion of the reaction, the resulting suspension was cooled at room temperature, diluted with water (10 mL), and extracted with ethyl acetate (5 × 100 mL). The combined organic layers were dried over anhydrous sodium sulfate, and after filtration of the solution, the solvent was evaporated under reduced pressure. The crude product was then purified by flash column chromatography (SiO_2_, *n*-hexane/ethyl acetate in gradient). A light yellow powder corresponding to **5** was obtained.

For 3,4-dimethoxy-9-oxo-9*H*-xanthene-1-carbaldehyde (**5**): Light yellow powder (1.33 g, 64% yield); m.p. > 330 °C. ^1^H-NMR (CDCl_3_, 300.13 MHz): *δ* = 11.21 (1H, s, H-1′), 8.31 (1H, dd, *J* = 8.0 and 1.7 Hz, H-8), 7.77 (1H, ddd, *J* = 8.5, 7.0, and 1.7 Hz, H-6), 7.60 (1H, dd, *J* = 8.5 and 0.8 Hz, H-5), 7.56 (1H, s, H-2), 7.43 (1H, ddd, *J* = 8.0, 7.0, and 0.8 Hz, H-7), 4.11 (3H, s, 4-OCH_3_), 4.07 (3H, s, 3-OCH_3_) ppm; ^13^C-NMR (CDCl_3_, 75.47 MHz): δ = 192.8 (C-1′), 177.9 (C-9), 156.3 (C-3), 155.5 (C-10a), 150.9 (C-4a), 140.7 (C-4), 135.1 (C-6), 133.5 (C-1), 126.7 (C-8), 124.5 (C-7), 121.9 (C-8a), 117.8 (C-5), 116.1 (C-9a), 108.5 (C-2), 61.8 (4-OCH_3_), 56.6 (3-OCH_3_) ppm. HRMS (ESI^+^): *m/z* [C_16_H_12_O_5_ + H]^+^ calcd. for [C_16_H_13_O_5_]: 285.07575; found: 285.07598; [C_16_H_12_O_5_ + Na]^−^ calcd. for [C_16_H_12_NaO_5_]: 307.05769; found 307.05750.

#### 3.1.4. Synthesis of 1-(hydroxymethyl)-3,4-dimethoxy-9*H*-xanthen-9-one (**29**)

Xanthone **29** (52 mg, 52% yield) was synthesized from xanthone **5** and characterized according to the previously described procedure [36].

#### 3.1.5. General Procedure for the Synthesis of Xanthones (**30**–**40**)

Xanthone **5** (40 mg, 0.141 mmol) was dissolved in tetrahydrofuran (3 mL) and the appropriate amine precursors **9**–**19** (0.197 mmol) were added to the solution under N_2_ gas. After the addition of a 4.0 equimolar quantity of sodium triacetoxyborohydride (119 mg, 0.563 mmol), the resulting mixture was stirred at room temperature for 30 min. Subsequently, a 3.0 equimolar quantity of acetic acid (24 μL 0.423 mmol) was added to the solution, The reaction was carried out at room temperature for 12 h under N_2_ gas. For monitoring the synthesis of xanthones **30**–**40** by TLC, two chromatographic systems were employed: (i) *n*-hexane/ethyl acetate (7:3) and methanol/triethylamine (10:0.1) for xanthones **30**–**36**; (ii) *n*-hexane/ethyl acetate (7:3) and chloroform/acetone/triethylamine (5:5:0.1) for xanthones **37**–**40**. Once completed of the reaction, three distinct work-up approaches were employed.

(a) For xanthones **30**–**32**: the solvent was evaporated under reduced pressure and the resulting crude product was basified with sodium hydroxide (10 mL, 5%, m/v) and extracted with chloroform (3 × 10 mL). The combined organic phases were then acidified and extracted with hydrochloric acid 5 M (3 × 30 mL). Afterwards, the aqueous phases were gathered and then basified with sodium hydroxide (100 mL, 20%, *m/v*). The crude product was extracted successively with chloroform (3 × 100 mL). The organic layers were gathered and dried over anhydrous sodium sulfate. After filtration, the solution was concentrated under reduced pressure, and the resulting crude product was purified by flash cartridge chromatography (SiO_2_, chloroform/acetone/ammonium hydroxide, 90:10:1) to afford the xanthones 30, 31, and 32 as pure compounds.

For 3,4-dimethoxy-1-(((2-morpholinoethyl)amino)methyl)-9*H*-xanthen-9-one (**30**): Green powder (35 mg, 62% yield); m.p. 125–126 °C. ^1^H-NMR (CDCl_3_, 300.13 MHz): *δ* = 8.25 (1H, dd, *J* = 8.0 and 1.6 Hz, H-8), 7.72 (1H, ddd, *J* = 8.1, 7.0, and 1.6 Hz, H-6), 7.57 (1H, dd, *J* = 8.4 and 0.8 Hz, H-5), 7.38 (1H, ddd, *J* = 8.0, 7.1, and 0.9 Hz, H-7), 7.09 (1H, s, H-2), 4.34 (2H, s, H-1′); 4.04 (3H, s, 4-OCH_3_), 4.01 (3H, s, 3-OCH_3_), 3.66 (4H, t, *J* = 4.6 Hz, H-5′), 2.90 (2H, t, *J* = 6.1 Hz, H-2′), 2.60 (2H, t, *J* = 6.1 Hz, H-3′), 2.47 (4H, t, *J* = 4.3 Hz, H-4′) ppm; ^13^C-NMR (CDCl_3_, 75.47 MHz): δ = 178.0 (C-9), 156.5 (C-3), 155.3 (C-10a), 152.1 (C-4a), 137.0 (C-1), 136.0 (C-4), 134.6 (C-6), 126.5 (C-8), 124.1 (C-7), 122.1 (C-8a), 117.7 (C-5), 114.6 (C-9a), 111.6 (C-2), 67.0 (C-5′), 61.5 (4-OCH_3_), 57.4 (C-2′), 56.4 (3-OCH_3_), 53.7 (C-1′), 53.5 (C-4′), 45.3 (C-3′) ppm. HRMS (ESI^+^): *m/z* [C_22_H_26_N_2_O_5_ + H]^+^ calcd. for [C_22_H_27_N_2_O_5_]: 399.19145; found: 399.19060.

For 1-(((3-(dimethylamino)propyl)(methyl)amino)methyl)-3,4-dimethoxy-9*H*-xanthen-9-one (**31**): White powder (35 mg, 65% yield); m.p. 84–85 °C. ^1^H-NMR (CDCl_3_, 300.13 MHz): δ = 8.25 (1H, dd, *J* = 8.0 and 1.6 Hz, H-8), 7.69 (1H, ddd, *J* = 8.5, 7.0, and 1.6 Hz, H-6), 7.54 (1H, dd, *J*= 7.4 and 0.7, H-5), 7.53 (1H, s, H-2), 7.35 (1H, ddd, *J* = 8.0, 7.1, and 1.0 Hz, H-7), 4.36 (2H, s, H-1′); 4.05 (3H, s, 4-OCH_3_), 4.02 (3H, s, 3-OCH_3_), 2.65 (2H, t, *J* = 7.2 Hz, H-3′), 2.44 (2H, t, *J* = 7.5 Hz, H-5′), 2.39 (3H, s, H-2′), 2.28 (6H, s, H-6′), 1.82 (2H, m, *J* = 7.3 Hz, H-4′) ppm; ^13^C-NMR (CDCl_3_, 75.47 MHz): δ = 178.1 (C-9), 156.6 (C-3), 155.2 (C-10a), 151.9 (C-4a), 135.2 (C-1), 135.0 (C-4), 134.2 (C-6), 126.6 (C-8), 123.9 (C-7), 122.4 (C-8a), 117.6 (C-5), 114.5 (C-9a), 109.0 (C-2), 61.5 (4-OCH_3_), 60.1 (C-1′), 57.8 (C-5′), 56.4 (3-OCH_3_), 56.1 (C-3′), 45.3 (C-6′), 42.5 (C-2′), 25.3 (C-4′) ppm. HRMS (ESI^+^): *m/z* [C_22_H_28_N_2_O_4_ + H]^+^ calcd. for [C_22_H_29_N_2_O_4_]: 385.21218; found: 385.21139.

For 1-((4-(2-hydroxyethyl)piperazin-1-yl)methyl)-3,4-dimethoxy-9*H*-xanthen-9-one (**32**): Yellow powder (40 mg, 71% yield); m.p. 129–130 °C. ^1^H-NMR (CDCl_3_, 300.13 MHz): *δ* = 8.25 (1H, dd, *J* = 8.0 and 1.6 Hz, H-8), 7.68 (1H, ddd, *J* = 8.5, 7.0, and 1.6 Hz, H-6), 7.53 (1H, dd, *J* = 8.4 and 0.7 Hz, H-5), 7.48 (1H, s, H-2), 7.35 (1H, ddd, *J* = 8.0, 7.1, and 1.0 Hz, H-7), 4.32 (2H, s, H-1′), 4.03 (3H, s, 4-OCH_3_), 4.01 (3H, s, 3-OCH_3_), 3.63 (2H, t, *J* = 5.4 Hz), 2.58 (2H, t, *J* = 5.4 Hz) ppm; ^13^C-NMR (CDCl_3_, 75.47 MHz): *δ* = 178.1 (C-9), 156.5 (C-3), 155.2 (C-10a), 152.0 (C-4a), 138.8 (C-1), 134.8 (C-4), 134.2 (C-6), 126.6 (C-8), 123.9 (C-7), 122.5 (C-8a), 117.5 (C-5), 114.6 (C-9a), 108.2 (C-2), 61.5 (4-OCH_3_), 60.2 (C-1′), 59.3 (C-5′), 57.7 (C-4′), 56.2 (3-OCH_3_), 53.5 (C-3′), 53.2 (C-2′) ppm. HRMS (ESI^+^): *m/z* [C_22_H_26_N_2_O_5_ + H]^+^ calcd. for [C_22_H_27_N_2_O_5_]: 399.19145; found: 399.19059.

(b) For xanthones **33**–**36**: the solvent was evaporated under reduced pressure and the crude product was extracted with chloroform (3 × 10 mL). The organic phases were gathered, dried over anhydrous sodium sulfate, and after filtration, the resulting solution was concentrated under reduced pressure. Then, a solid phase extraction using a cation exchange cartridge Discovery^®^ DSC-SCX was applied to further purify the crude product. Initially, an activation of the cartridge with methanol (100 mL) was carried out followed by loading the cartridge with the sample. Then, the elution was performed with the following solvents/solutions: (i) dichloromethane/methanol (5:5), (ii) methanol, and (iii) ammonium hydroxide/methanol (2%, *v/v*). The fractions obtained from the elution with ammonium hydroxide/methanol (2%, *v/v*) were gathered and the solvent was evaporated under reduced pressure. A flash cartridge chromatography with chloroform/acetone/ammonium hydroxide (90:10:1) was also performed to obtain the xanthones **33**, **34**, **35**, and **36** as pure compounds.

For 1-(((2-(diethylamino)ethyl)amino)methyl)-3,4-dimethoxy-9*H*-xanthen-9-one (**33**): Green powder (28 mg, 52% yield); m.p. 68–69 °C. ^1^H-NMR (CDCl_3_, 300.13 MHz): *δ* = 8.27 (1H, dd, *J* = 8.0 and 1.4 Hz, H-8), 7.71 (1H, ddd, *J* = 8.5, 7.0, and 1.6 Hz, H-6), 7.56 (1H, dd, *J* = 8.4 and 0.7 Hz, H-5), 7.37 (1H, ddd, *J* = 8.0, 7.1, and 1.0 Hz, H-7), 7.14 (1H, s, H-2), 4.39 (2H, s, H-1′); 4.05 (3H, s, 4-OCH_3_), 4.01 (3H, s, 3-OCH_3_), 2.87 (2H, t, *J* = 6.4 Hz, H-2′), 2.68 (2H, t, *J* = 6.4 Hz, H-3′), 2.55 (4H, m, *J* = 7.1 Hz, H-4′), 1.01 (6H, t, *J* = 7.1 Hz, H-5′) ppm; ^13^C-NMR (CDCl_3_, 75.47 MHz): δ = 178.1 (C-9), 156.5 (C-3), 155.3 (C-10a), 152.1 (C-4a), 137.4 (C-1), 135.8 (C-4), 134.5 (C-6), 126.6 (C-8), 124.0 (C-7), 122.2 (C-8a), 117.7 (C-5), 114.5 (C-9a), 111.1 (C-2), 61.5 (4-OCH_3_), 56.4 (3-OCH_3_), 53.4 (C-3′), 52.4 (C-1′), 46.9 (C-4′), 46.8 (C-2′), 11.6 (C-5′) ppm. HRMS (ESI^+^): *m/z* [C_22_H_28_N_2_O_4_ + H]^+^ calcd. for [C_22_H_29_N_2_O_4_]: 385.21218; found: 385.21150.

For 4-((3,4-dimethoxy-9-oxo-9*H*-xanthen-1-yl)methyl)piperazin-2-one (**34**): Orange powder (20 mg, 39% yield); m.p. 245–247 °C. ^1^H-NMR (DMSO-*d*_6_, 300.13 MHz): *δ* = 8.13 (1H, dd, *J* = 8.0 and 1.5 Hz, H-8), 7.82 (1H, ddd, *J* = 8.5, 7.0, and 1.6 Hz, H-6), 7.65 (1H, dd, *J* = 8.4 and 0.6, H-5), 7.45 (1H, ddd, *J* = 8.0, 7.1, and 1.0 Hz, H-7), 7.36 (1H, s, H-2), 4.24 (2H, s, H-1′); 3.98 (3H, s, 4-OCH_3_), 3.90 (3H, s, 3-OCH_3_), 3.20 (2H, t, *J* = 5.3 Hz, H-5′), 3.13 (2H, s, H-2′), 2.66 (2H, t, *J* = 5.3 Hz, H-4′) ppm; ^13^C-NMR (DMSO-*d*_6_, 75.47 MHz): *δ* = 176.7 (C-9), 168.0 (C-3′), 156.2 (C-3), 154.6 (C-10a), 151.8 (C-4a), 136.9 (C-1), 135.0 (C-4), 134.7 (C-6), 126.1 (C-8), 124.3 (C-7), 121.8 (C-8a), 117.7 (C-5), 113.7 (C-9a), 109.2 (C-2), 60.9 (4-OCH_3_), 58.5 (C-1′), 57.2 (C-5′), 56.3 (3-OCH_3_), 48.7 (C-2′), 30.7 (C-4′) ppm. HRMS (ESI^+^): *m/z* [C_20_H_20_N_2_O_5_ + H]^+^ calcd. for [C_20_H_21_N_2_O_5_]: 369.14450; found: 369.14369.

For 1-((5-amino-3,4-dihydroisoquinolin-2(1*H*)-yl)methyl)-3,4-dimethoxy-9*H*-xanthen-9-one (**35**): White powder (23 mg, 39% yield); m.p. 171–172 °C. ^1^H-NMR (CDCl_3_, 300.13 MHz): *δ* = 8.26 (1H, dd, *J* = 8.0 and 1.4 Hz, H-8), 7.69 (1H, ddd, *J* = 8.5, 7.0, and 1.6 Hz, H-6), 7.57 (1H, s, H-2), 7.54 (1H, dd, *J* = 8.4 and 0.7 Hz, H-5), 7.35 (1H, ddd, *J* = 8.0, 7.1, and 1.0 Hz, H-7), 6.99 (1H, t, *J* = 7.7 Hz, H-7′), 6.57 (1H, d, *J* = 7.8 Hz, H-6′), 6.53 (1H, d, *J* = 7.6 Hz, H-8′), 4.48 (2H, s, H-1′); 4.02 (3H, s, 4-OCH_3_), 3.97 (3H, s, 3-OCH_3_), 3.81 (2H, s, H-10′), 2.93 (2H, t, *J* = 5.9 Hz, H-2′), 2.66 (2H, t, *J* = 5.9 Hz, H-3′) ppm; ^13^C-NMR (CDCl_3_, 75.47 MHz): *δ* = 178.2 (C-9), 156.7 (C-3), 155.2 (C-10a), 151.9 (C-4a), 144.1 (C-5′), 139.3 (C-1), 136.3 (C-9′), 134.8 (C-4), 134.2 (C-6), 126.6 (C-8), 126.4 (C-7′), 123.8 (C-7), 122.5 (C-8a), 119.5 (C-4′), 117.6 (C-5), 117.1 (C-8′), 114.5 (C-9a), 112.7 (C-6′), 108.0 (C-2), 61.5 (4-OCH_3_), 60.1 (C-1′), 56.7 (C-10′), 56.3 (3-OCH_3_), 51.0 (C-2′), 24.9 (C-3′) ppm. HRMS (ESI^+^): *m/z* [C_25_H_24_N_2_O_4_ + H]^+^ calcd. for [C_25_H_25_N_2_O_4_]: 417.18088; found: 417.18018.

For 3,4-dimethoxy-1-(piperidin-1-ylmethyl)-9*H*-xanthen-9-one (**36**): Yellow powder (31 mg, 62% yield); m.p. 108-109 °C. ^1^H-NMR (CDCl_3_, 300.13 MHz): *δ* = 8.25 (1H, dd, *J* = 8.0 and 1.6 Hz, H-8), 7.66 (1H, ddd, *J* = 8.5, 8.0, and 1.6 Hz, H-6), 7.56 (1H, s, H-2), 7.52 (1H, dd, *J* = 8.3 and 0.7 Hz, H-5), 7.33 (1H, ddd, *J* = 7.9, 7.1, and 0.9 Hz, H-7), 4.26 (2H, s, H-1′); 4.03 (3H, s, 4-OCH_3_), 4.01 (3H, s, 3-OCH_3_), 2.57 (4H, t, *J* = 4.6 Hz, H-2′), 1.65 (4H, m, *J* = 5.5 Hz, H-3′), 1,50 (2H, m, *J* = 5.5 Hz, H-4′) ppm; ^13^C-NMR (CDCl_3_, 75.47 MHz) ppm: *δ* = 177.5 (C-9), 156.1 (C-3), 154.7 (C-10a), 151.3 (C-4a), 138.9 (C-4), 134.2 (C-1), 133.8 (C-6), 126.0 (C-8), 123.3 (C-7), 121.9 (C-8a), 117.1 (C-5), 113.9 (C-9a), 107.8 (C-2), 60.8 (4-OCH_3_), 56.1 (3-OCH_3_), 55.7 (C-2′), 54.3 (C-1′), 25.7 (C-3′), 23.8 (C-4′) ppm. HRMS (ESI^+^): *m/z* [C_21_H_23_NO_4_ + H]^+^ calcd. for [C_21_H_24_NO_4_]: 354.16998; found: 354.16939.

(c) For xanthones **37**–**40**: the solvent was evaporated under reduced pressure and the resulting crude product was basified with sodium hydroxide (10 mL, 5%, *m/v*) and extracted with chloroform (3 × 10 mL). The combined organic phases were then acidified and extracted with hydrochloric acid 5 M (3 × 30 mL). The resulting organic phase was dried over anhydrous sodium sulfate and, after filtration, the solution was concentrated under reduced pressure. Then, a solid phase extraction using a cation exchange cartridge Discovery^®^ DSC-SCX was applied to further purify the crude product. Initially, an activation of the cartridge with methanol (100 mL) was carried out followed by loading the cartridge with the sample. Then, the elution was performed with the following solvents/solutions: (i) dichloromethane/methanol (5:5), (ii) methanol, and (iii) ammonium hydroxide/methanol (2%, *v/v*). The fractions obtained from the elution with ammonium hydroxide/methanol (2%, *v/v*) were gathered and the solvent was evaporated under reduced pressure to furnish the xanthones **37**, **38**, **39**, and **40** as pure compounds.

For 1-(((4-fluorobenzyl)amino)methyl)-3,4-dimethoxy-9*H*-xanthen-9-one (**37**): White powder, (34 mg, 61% yield); m.p. 163–165 °C. ^1^H-NMR (CDCl_3_, 300.13 MHz): *δ* = 8.28 (1H, dd, *J* = 8.0 and 1.6 Hz, H-8), 7.71 (1H, ddd, *J* = 8.5, 7.0, and 1.6 Hz, H-6), 7.55 (1H, dd, *J* = 8.5 and 0.8 Hz, H-5), 7.40 (1H, ddd, *J* = 8.0, 7.0, and 0.8 Hz, H-7), 7.36 (2H, m, H-5′), 7.01 (2H, m, H-4′), 6.96 (1H, s, H-2), 4.30 (2H, s, H-1′), 4.01 (3H, s, 4-OCH_3_), 4.01 (3H, s, 3-OCH_3_), 3.85 (2H, s, H-2′) ppm; ^13^C-NMR (CDCl_3_, 75.47 MHz): *δ =* 178.1 (C-9), 160.3 (C-6′), 156.3 (C-3), 155.3 (C-10a), 152.2 (C-4a), 138.6 (C-4), 135.9 (C-3′), 135.7 (C-1), 134.5 (C-6), 129.8 (C-5′), 126.7 (C-8), 124.0 (C-7), 122.2 (C-8a), 117.7 (C-5), 115.3 (C-4′), 114.7 (C-9a), 110.9 (C-2), 61.5 (4-OCH_3_), 56.3 (3-OCH_3_), 53.3 (C-1′), 52.5 (C-2′) ppm. HRMS (ESI^+^): *m/z* [C_23_H_20_FNO_4_ + H]^+^ calcd. for [C_23_H_21_FNO_4_]: 394.14491; found: 394.14410.

For 1-(((4-chlorobenzyl)amino)methyl)-3,4-dimethoxy-9*H*-xanthen-9-one (**38**): Light yellow powder (35 mg, 61% yield); m.p. 162–163 °C. ^1^H-NMR (CDCl_3_, 300.13 MHz): *δ* = 8.25 (1H, dd, *J* = 8.0 and 1.6 Hz, H-8), 7.79 (1H, ddd, *J* = 8.5, 7.0, and 1.6 Hz, H-6), 7.66 (2H, m, H-5′), 7.60 (1H, dd, *J* = 8.5 and 0.9 Hz, H-5), 7.43 (1H, ddd, *J* = 8.0, 7.0, and 0.9 Hz, H-7), 7.39 (1H, s, H-2), 7.38 (2H, m, H-4′), 4.53 (2H, s, H-2′); 4.33 (2H, s, H-1′), 4.07 (3H, s, 4-OCH_3_), 4.01 (3H, s, 3-OCH_3_) ppm; ^13^C-NMR (CDCl_3_, 75.47 MHz): *δ* = 179.6 (C-9), 157.0 (C-3), 155.5 (C-10a), 152.0 (C-4a), 137.7 (C-4), 135.8 (C-1), 135.6 (C-6), 131.7 (C-5′), 129.5 (C-4′), 129.4 (C-3′), 127.3 (C-6′), 126.6 (C-8), 124.6 (C-7), 121.3 (C-8a), 117.9 (C-5), 115.5 (C-2), 114.8 (C-9a), 61.6 (4-OCH_3_), 57.1 (3-OCH_3_), 51.3 (C-1′), 50.7 (C-2′) ppm. HRMS (ESI^+^): *m/z* [C_23_H_20_ClNO_4_ + H]^+^ calcd. for [C_23_H_21_ClNO_4_]: 410.11536; found: 410.11466.

For (*R*)-1-(((1-(4-chlorophenyl)ethyl)amino)methyl)-3,4-dimethoxy-9*H*-xanthen-9-one (**39**): White powder (39 mg, 65% yield); m.p. 163–164 °C. ^1^H-NMR (CDCl_3_, 300.13 MHz): δ = 8.26 (1H, dd, *J* = 8.0 and 1.6 Hz, H-8), 7.79 (1H, ddd, *J* = 8.5, 7.0, and 1.6 Hz, H-6), 7.70 (2H, m, H-6′), 7.59 (1H, dd, *J* = 8.5 and 0.8 Hz, H-5), 7.43 (1H, ddd, *J* = 8.0, 7.0, and 0.8 Hz, H-7), 7.42 (2H, m, H-5′), 7.25 (1H, s, H-2), 4.54 (2H, s, H-1′); 4.27 (1H, q, *J* = 6.8 Hz, H-1′), 4.04 (3H, s, 4-OCH_3_), 4.01 (3H, s, 3-OCH_3_), 1.83 (3H, d, *J* = 6.8 Hz, H-3′) ppm; ^13^C-NMR (CDCl_3_, 75.47 MHz): *δ* = 179.6 (C-9), 156.8 (C-3), 155.4 (C-10a), 151.9 (C-4a), 137.6 (C-4), 135.5 (C-6), 134.5 (C-1), 131.8 (C-7′), 129.8 (C-6′), 129.5 (C-5′), 127.3 (C-4′), 126.6 (C-8), 124.6 (C-7), 121.3 (C-8a), 117.9 (C-5), 115.6 (C-2), 114.9 (C-9a), 61.6 (4-OCH_3_), 58.9 (C-2′), 57.0 (3-OCH_3_), 50.2 (C-1′), 20.2 (C-3′) ppm. HRMS (ESI^+^): *m/z* [C_24_H_22_ClNO_4_ + H]^+^ calcd. for [C_24_H_23_ClNO_4_]: 424.13101; found: 424.13018.

For 1-(((4-bromobenzyl)amino)methyl)-3,4-dimethoxy-9*H*-xanthen-9-one (**40**): White powder (47 mg, 73% yield); m.p. 161–162 °C. ^1^H-NMR (CDCl_3_, 300.13 MHz): *δ* = 8.27 (1H, dd, *J* = 8.0 and 1.6 Hz, H-8), 7.73 (1H, ddd, *J* = 8.5, 7.0, and 1.6 Hz, H-6), 7.58 (1H, dd, *J* = 8.5 and 0.8 Hz, H-5), 7.48 (2H, m, H-5′), 7.39 (2H, m, H-4′), 7.38 (1H, ddd, *J* = 8.0, 7.0, and 0.8 Hz, H-7), 7.08 (1H, s, H-2), 4.31 (2H, s, H-1′); 4.01 (3H, s, 4-OCH_3_), 4.01 (3H, s, 3-OCH_3_), 3.88 (2H, s, H-2′) ppm; ^13^C-NMR (CDCl_3_, 75.47 MHz): *δ* = 178.6 (C-9), 156.5 (C-3), 155.3 (C-10a), 152.2 (C-4a), 136.4 (C-4), 136.2 (C-1), 134.9 (C-6), 131.8 (C-5′), 130.7 (C-4′), 128.4 (C-3′), 126.6 (C-8), 124.2 (C-7), 121.9 (C-8a), 121.8 (C-6′), 117.7 (C-5), 114.7 (C-9a), 112.5 (C-2), 61.6 (4-OCH_3_), 56.7 (3-OCH_3_), 52.7 (C-1′), 52.0 (C-2′) ppm. HRMS (ESI^+^): *m/z* [C_23_H_20_BrNO_4_ + H]^+^ calcd. for [C_23_H_21_BrNO_4_]: 454.06485; found: 454.06385.

### 3.2. Biological Evaluation

#### 3.2.1. Yeast Strains and Growth Conditions

For the yeast assay, the previously obtained *Saccharomyces cerevisiae* cells co-expressing human p53 or human MDM2, as well as yeast control (transformed with empty vectors) [30,31], were used. To induce expression of human proteins, yeast cells were grown in selective induction medium with 2% (*w/w*) galactose and 2% (*w/w*) raffinose, in the presence of 1–20 μM of xanthones **30**–**40** or 0.1% DMSO, at 30 °C under continuous orbital shaking (200 rpm) for approximately 42 h, as described [30,31]. Yeast cell growth was analyzed by counting the number of colony-forming units (CFU) after 2 days incubation at 30 °C on Sabouraud Dextrose Agar plates from Liofilchem (Frilabo, Porto, Portugal). Results were estimated considering 100% growth as the number of CFU obtained with untreated yeast co-expressing p53 and MDM2.

#### 3.2.2. Human Cancer Cell Lines and Culture Conditions

Human colon adenocarcinoma HCT116 cell lines expressing wt p53 (HCT116 p53^+/+^) and its p53-null isogenic derivative (HCT116 p53^−/−^) were provided by B. Vogelstein (The Johns Hopkins Kimmel Cancer Center, Baltimore, MD, USA). Human liver HepG2 carcinoma cell lines were purchased from ATCC (Rockville, MD, USA). Cancer cells were cultured in RPMI-1640 medium with ultraglutamine (Lonza, VWR, Carnaxide, Portugal), and supplemented with 10% fetal bovine serum (FBS; Merck Millipore, VWR). Cells were maintained at 37 °C in a humidified atmosphere of 5% CO_2_.

#### 3.2.3. Cell Proliferation Assay

Cell proliferation was determined using the SRB assay, as described [30,31]. Briefly, cells were seeded in 96-well plates at 5.0 × 10^3^ cells/well for HCT116 p53^+/+^ cells and 4.0 × 10^3^ cells/well for HepG2 and 24 h later treated with serial dilutions (3.13–50 µM) of xanthone **37** or 0.25% DMSO for 48 h. IC_50_ values were determined from the concentration-response curves.

#### 3.2.4. Colony Formation Assay

HCT116 p53^+/+^ and HCT116 p53^−/−^ cells were seeded in a 6-well plate at 1.0 × 10^3^ cells/well and treated with 3, 4, 5, 7, and 10 µM of xanthone **37** or DMSO at the time of seeding. After 11 days incubation, colonies were fixed using 10% of methanol and 10% acetic acid for 10 min and stained with 0.5% crystal violet (Sigma-Aldrich) in 1:1 methanol/water for 15 min. Colonies with more than 20 cells were counted.

#### 3.2.5. Cell Cycle Analysis

HCT116 p53^+/+^ and HCT116 p53^−/−^ cells were seeded in a 6-well plate at 1.5 × 10^5^ cells/well for 24 h and treated with 10 and 20 µM of xanthone **37** or DMSO for 24 h. Cells were collected, fixed, and stained with propidium iodide (Fluka, Sigma-Aldrich). Cell cycle was analyzed by flow cytometry, as described [45].

#### 3.2.6. Western Blot Analysis

HCT116 p53^+/+^ were seeded in a 6-well plate at 1.5 × 10^5^ cells/well for 24 h and treated with 10 and 20 µM of xanthone **37** or DMSO for 24 h. Total protein extracts of cancer cells were obtained and analyzed by Western blot, as described [45]. Membranes were probed with a mouse monoclonal anti-p53 (DO-1), anti-MDM2 (D-12), or with a rabbit polyclonal anti-p21 (C-19), followed by an anti-mouse or anti-rabbit horseradish-peroxidase (HRP)-conjugated secondary antibody. For loading control, membranes were stripped and re-probed with mouse monoclonal anti-GAPDH (6C5). All antibodies were purchased from Santa Cruz Biotechnology (Frilabo, Porto, Portugal). The signal was detected with enhanced chemiluminescence (ECL) Amersham Kit from GE Healthcare (VWR) and with Molecular Imager^®^ ChemiDoc™ XRS+ System, using Image Lab™ software.

#### 3.2.7. Statistical Analysis

Data’s statistical analysis was performed using the Prism 7 program (GraphPad Software Inc., California, USA). Differences between means were tested for significance using one-way ANOVA with Tukey’s multiple comparisons test, two-way ANOVA with Sidak’s multiple comparisons test, and Student’s *t*-test (* *p* < 0.05; ** *p* < 0.01).

### 3.3. In Silico Studies onto MDM2

The chemical structures of gambogic acid (**2**), *α*-mangostin (**3**), 12-hydroxy-2,2-dimethyl-3,4-dihydro-2*H*,6*H*-pyrano[3,2-*b*]xanthen-6-one (**4**), and the xanthones **30**–**40** were drawn using ChemSketch (Advanced Chemistry Development, Inc. (ACD/Labs), Canada); the structure of known MDM2-p53 inhibitor nutlin-3A (**1**) was obtained from PubChem [53]. The three-dimensional (3D) structures of the xanthones and **1** were minimized using the ArgusLab version 4.0.1 software for Windows by Hamiltonian (quantum mechanics) using the Parameterized Model number 3 (PM3) semi-empirical method [54]. The 3D structure of MDM2 was extracted from Protein Data Bank (PDB id 4HG7) [50]. Docking simulations involving MDM2 and the small molecules were undertaken in AutoDock Vina (Scripps Research Institute, USA) [51]. AutoDock Vina considered the macromolecular target conformation as a rigid unit, while the ligands were allowed to be flexible and adaptable to the target. Vina explored the lowest binding affinity conformations and returned nine different conformations for each ligand. AutoDock Vina was run using an exhaustiveness of 8 and a grid box with the dimensions of 19.7 Å, 26.1 Å, and 14.1 Å, engulfing the binding cavity occupied by the crystallographic nutlin-3A (PDB id 4HG7). Ligand conformations and interactions with the binding cavity were visualized using PyMOL version 1.3 [55]. In order to validate the docking approach for the macromolecular target structure used, nutlin-3A was docked onto the binding cavity of MDM2 using AutoDock Vina, and the obtained docking conformation was compared to crystallographic nutlin-3A (PDB id 4HG7) using RMSD (not shown). Docking scores of xanthones and positive controls were compared.

## 4. Conclusions

An alternative and efficient strategy for the preparation of carbaldehydic xanthones was successfully developed. The use of a precursor with a methyl substituent at position C-1 revealed to be crucial for the optimization of the synthesis of **5**. The newly described approach required lower reaction times and enabled the preparation of xanthone **5** in a higher global yield. In this work, a series of eleven aminated xanthones (**30**–**40**) possessing a 3,4-dioxygenated pattern of substitution was efficiently constructed in moderate to good yields. From this group of compounds, xanthone **37** was identified for the first time as a putative p53-activating agent, using a yeast-based screening assay. Xanthone **37** was revealed to inhibit the growth of human HCT116 p53^+/+^ colon cancer cells, being that this effect is associated with cell cycle arrest through activation of the p53 pathway. Nevertheless, further studies are required to confirm the mechanism of action of **37**, which may lead to the identification of a novel xanthone derivative with promising antitumor activity.

These results demonstrated the potential usefulness of coupling amine-containing structural motifs of known MDM2-p53 disruptors into the 3,4-dioxygenated xanthone scaffold as a starting point for the design of novel and improved p53-activating agents with antitumor activity and drug-like properties.

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
