# Peer review of "Synthesis, Biological Evaluation, and In Silico Studies of Novel Aminated Xanthones as Potential p53-Activating Agents"

_molecules, 2019, doi:10.3390/molecules24101975_

Reviewer 1 Report

In this paper Sousa et al. describe the synthesis of some aminated 3,4-dioxygenated xanthones, substances known to possess antitumor activity against several human tumour cells.

Recent studies from the same group had previously shown that two 3,4-dioxygenated xanthones were MDM2-p53 inhibitor or Tap73activator. Based upon these results a small library of aminated xanthones was constructed by a molecular hybridization approach combining a 3,4-dimethoxy xanthone scaffold with structural motifs of known MDM2-p53 disruptors. Reductive amination was used to insert the amine moiety at C-1 of the parent xanthone.

As far as the the chemistry is concerned, an improvement of the route leading to the parent scaffold has been achieved.

A yeast-based assay, previously developed by the same group, the evaluation of the antitumor activity of the synthesized substances on human cancer cell lines as well as in silico docking studies were also carried out. Importantly, one of the synthesized compounds has shown the ability to inhibit the growth of a colon cancer cell line through activation of the p53 pathway.

In the introduction, however, authors should better explain what is meant by “chemical moieties” of MDM2-p53 inhibitors, shown in figure 2. In other words, not only the amine portions of these compounds are used for hybridization but also fluoro-, chloro- or bromo-phenyl moieties are transformed into amines and appended to the xanthone scaffold. This should be clearly explained. Also, the caption of figure 2 should be clearer.

In addition:

Line 114: replace “construction” with “planning or accomplishment”;

Line 155: 29 in place of 26?

Line 159: delete “a scope of”;

Line 573 and throughout the paper: replace “scope” with another word (collection?).

Therefore, overall, in my opinion this is a well-articulated and scientifically sound paper that deserves publication on Molecules.

Author Response

In the introduction, however, authors should better explain what is meant by “chemical moieties” of MDM2-p53 inhibitors, shown in figure 2. In other words, not only the amine portions of these compounds are used for hybridization but also fluoro-, chloro- or bromo-phenyl moieties are transformed into amines and appended to the xanthone scaffold. This should be clearly explained. Also, the caption of figure 2 should be clearer.

Answer: In the Fig.2 (revised manuscript version), we replaced the chemical structure of the compound 6 to a different one containing a fluorobenzyl-N portion. For the compound 7, we highlighted in red the chlorobenzyl-N portion in the revised manuscript version. We believe that these changes may clarify the reader about the implemented hybridization approach. Also, we included the chemical structures of the amine precursors selected for the reductive amination reaction with the xanthone 5.  

 Line 114: replace “construction” with “planning or accomplishment”;

Answer: We agree with the comment of reviewer #1 and the word “construction” was replaced by “planning”.

 Line 155: 29 in place of 26?

Answer: We agree with the comment of reviewer #1 and the number “26” was replaced by “29”.

 Line 159: delete “a scope of”;

Answer: We agree with the comment of reviewer #1 and the expression “a scope of” was deleted.

 Line 573 and throughout the paper: replace “scope” with another word (collection?).

Answer: We agree with the comment of the reviewer #1 and the word “scope” was deleted in line 159. Throughout the manuscript, the word “scope” was replaced by “set” in line 102 (submitted version) and by “series” in line 573 (submitted version).    

Reviewer 2 Report

In this manuscript, a series of new aminated xanthones were successfully synthesized, whose abilities to disrupt the MDM2-p53 interaction were evaluated, and in-silico studies were conducted as well. The manuscript is well written and organized, and sufficient data and adequate method description have been provided. Thus, I would recommend this paper to be published in Molecules.

Author Response

We are grateful to the reviewers for their constructive comments and feel that we have been able to addressed them, which resulted in a better manuscript.

Reviewer 3 Report

In general, the paper is well written except for a few grammatical errors.

Some questions arise as shown below:

    a.  p 5, line 159-"eleven laminated xanthines 27-37".  However, on page 12, line 354-355 it is indicated that "appropriate amines 18-28".  Should it read "aminated amino-xanthines"?  

    b.  On page 6, simple amines 18-28 are cited.   Some numbers need correction.

Why are there no melting points for the solids although listed as powders in the Experimental?

Author Response

Some questions arise as shown below:

    a.  p 5, line 159-"eleven laminated xanthines 27-37".  However, on page 12, line 354-355 it is indicated that "appropriate amines 18-28".  Should it read "aminated amino-xanthines"?  

Answer: In the submitted manuscript version, the amines 18-28 were used as precursors in the reaction of reductive amination with the xanthone 5. In the revised manuscript version, the chemical structures of these amine precursors are represented in Figure 2C. The numbering of the amine precursors 18-28 was changed to “9-19”. In addition, we suggested to replace the expression “appropriate amines 18-28” by “appropriate amine precursors 9-19” to differentiate the precursors (“simple amines”) and the final products (“aminated xanthones”).

     b.  On page 6, simple amines 18-28 are cited.  Some numbers need correction.

Answer: We followed the reviewer’s advice and, in the revised manuscript version, the numbering of the amine precursors 18-28 and the aminated xanthones 27-37 (comment a.) was replaced by “9-19” and “30-40”, respectively.

 Why are there no melting points for the solids although listed as powders in the Experimental?

Answer: In the Material and Methods section, the melting points (m.p.) were included as highlighted (in red) for all the synthesized solid compounds.

 We are grateful to the reviewers for their constructive comments and feel that we have been able to addressed them, which resulted in a better manuscript.